# WebCanvas: Benchmarking Web Agents in Online Environments

## Abstract

For web agents to be practically useful, they need to generalize to the ever changing web environment — UI updates, page content updates, etc. Unfortunately, most traditional benchmarks only capture a static state of the web page. We introduce **WebCanvas**, an innovative online evaluation framework for web agents designed to address the dynamic nature of web interactions. Web-Canvas contains three main components supporting realistic assessments: (1) A key-node-based evaluation metric, which stably capture critical actions or states necessary for task completions while disregarding noises caused by insignificant events or changed web-elements; (2) A benchmark dataset called Mind2Web-Live, a refined version of original Mind2Web static dataset containing **542** tasks with **2439** intermediate evaluation states; (3) Lightweight and generalizable annotation tools and testing pipelines, which allows us to maintain the high-quality, up-to-date dataset and automatically detection shifts in live action sequences. Despite the advancements, best-performing model achieves only a 23.1% task success rate, highlighting substantial room for improvement in future work.

## 1. Introduction

Unlike text generation tasks that leverages built-in model knowledge (Hendrycks et al., 2020; Chen et al., 2021; Cobbe et al., 2021), agents require environmental observations and feedback for context. Thus, dynamic, real-world environments are essential for agent evaluation. The World Wide Web itself emerges as the most extensive arena for the assessment of agents, offering an unparalleled complexity for environmental interaction. However, the rapid evolution of the web environment, driven by technological advancements and evolving design trends, introduces significant data distribution shifts over time. Figure 1 shows three common patterns of web tasks shift over time. in the course of our study, it was discovered that **10%** of the tasks annotated within the renowned benchmark dataset Mind2Web (Deng et al., 2024) in May 2023 have become totally obsolete, and

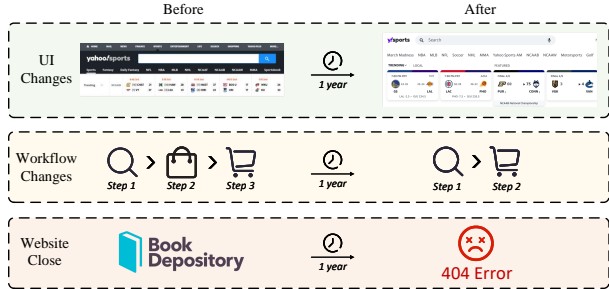

Figure 1: Illustration of how web tasks change over time due to various factors, placing challenges in building benchmarks in online web environment where changes can significantly alter task pathways and outcomes.

an additional **38%** have undergone modifications in their completion logic and trajectories at the time of May 2024. This data shift is fine as they are valid in the original setup, but it has the potential to bring gaps between the offline and online evaluation and development processes of real-world web agents. Meanwhile, the accumulated knowledge and training data of static websites leads to the saturation of existing benchmarks, making it increasingly difficult to compare models and reasoning frameworks fairly and rigorously. We found model trained in 2023 have already had performance discrepancy between offline and online evaluation as compared with close-sourced model such as GPT-3.5 (Ouyang et al., 2022) and GPT-4 (Achiam et al., 2023), which is discussed in §J.3.

Motivated by these demands, previous works have attempted human evaluation of web agents performance in online environment (Zheng et al., 2024; He et al., 2024), yet there lacks an objective, quantitative and automated method for evaluation. Thus, we introduce **WebCanvas**, a dynamic and real-time framework designed for online evaluation of web agents. **WebCanvas** is built on the following principles:

**1. Key nodes annotation provides in-progress evaluation.** In the dynamic and open environment of online benchmarks, traditional evaluation methods that focus solely on action prediction accuracy (Deng et al., 2024; Zheng et al., 2024) and the final state achievement (Zhou et al., 2023; Mialon et al., 2023) could not effectively adapt to the complexities

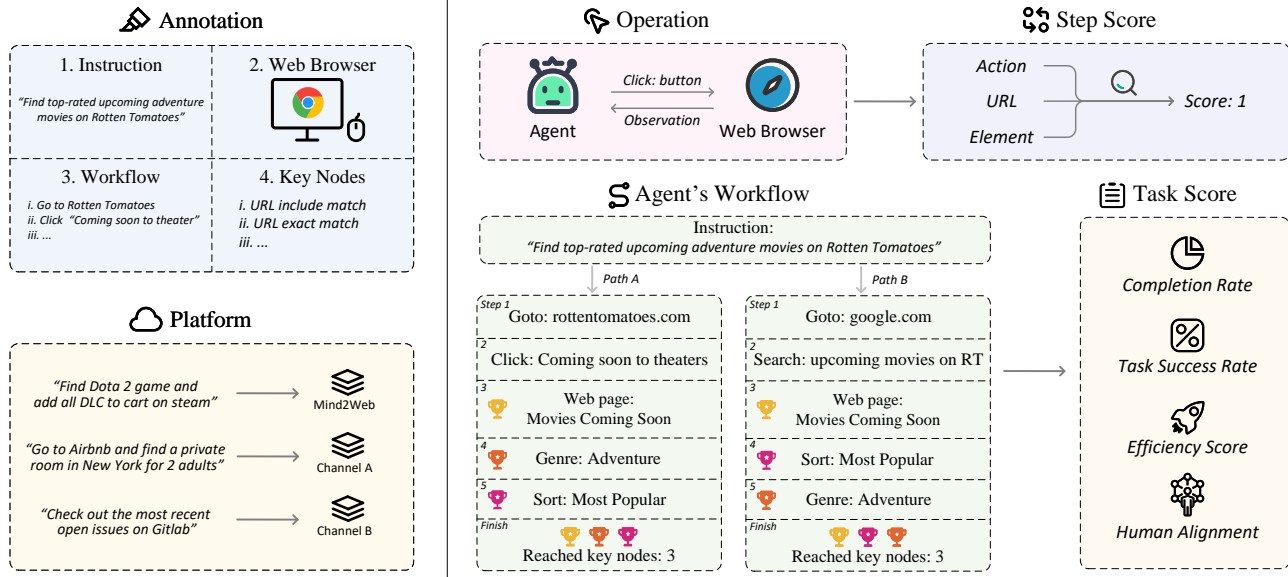

Figure 2: WebCanvas framework. The left side depicts the annotation process addressing each task, while the right side demonstrates the evaluation process during inference time, which involves collection of predicted actions, URLs, and elements targeted for interaction in online web environment, allowing for dynamic assessment. The framework accounts for the non-uniqueness of paths in online web interactions, with "Trophies" representing step scores earned upon successfully reaching each key node. The process of data maintenance related to these activities is detailed in §2.4.2.

of real-world web navigation. To address this gap, we introduce a novel concept termed "key nodes" — essential milestones that any task process must traverse, irrespective of the path taken.

Implementing these key nodes allows for a detailed, step-by-step analysis of agent behaviors during task execution, thereby enhancing our understanding of the agent's decision-making strengths and weaknesses. Based on the key nodes definition and evaluation metrics construction, we developed the **Mind2Web-Live** dataset as a foundation benchmark for the community. We sampled and annotated tasks from Mind2Web (Deng et al., 2024), maintaining a collection of 542 web agent tasks, including 2439 intermediate evaluation states.

**2. Easy to Scale in Online Web Environment.** WebCanvas supports recording and annotation of web agent tasks and their corresponding evaluation states through an advanced recording browser plugin with transparent data access. Furthermore, we have open-sourced an agent reasoning framework that enhances the integration and customization of various agent modules for online web tasks. This initiative provided guidelines and toolkits for the community to effectively scale data for online evaluation within real-world settings in their own scenario.

**3. Regular maintenance under optimized cost.** Online environment is continuously evolving, making maintain-

ing data validity a challenge. To address this, WebCanvas employs an efficient maintenance strategy with scheduled monitoring and automated alerts that quickly identify action sequences and key nodes validity. When data shifts occur, our test report with error messages guide data owner through quick and effective data corrections. This approach allows us to dynamically adjust our evaluation sets in response to real-time changes in web content with acceptable cost.

Comparative experiments across different models and settings in online web environment found that GPT-4-turbo achieved the best completion rate of **48.8%** and task success rate of **23.1%** on **Mind2Web-Live** test set, leaving substantial room for improvement in future work. We also conducted a rigorous analysis to dissect various factors influencing online agent performance and quantitatively explored the distinctions between WebCanvas and previous evaluation methods for web agent.

## 2. WebCanvas: An Online Evaluation Framework for Web Agents

### 2.1. Problem formulation

The real-world web environment can be formulated as: $(\mathcal{S}, \mathcal{A}, \mathcal{T}, \mathcal{O})$ with state space $\mathcal{S}$, action space $\mathcal{A}$(Table 7), deterministic transition function $\mathcal{T} : \mathcal{S} \times \mathcal{A} \longrightarrow \mathcal{S}$ and a state observation space $\mathcal{O}$(§3.1). Given a task instruction

$i$, current observation $o_t \in \mathcal{O}$ and the action history $a_1^{t-1}$, an agent issues an action $a_t \in \mathcal{A}$. Consequently, after the execution of the action, the environment transitions to a new state $s_{t+1} \in \mathcal{S}$, and the corresponding observation updates to $o_{t+1} \in \mathcal{O}$. To measure the completion of tasks, we have defined key nodes and evaluation metrics, which are elaborated in §2.2 and §2.3.

## 2.2. Key nodes

The concept of "key nodes" is one of the pivotal ideas in our work. Key nodes refer to indispensable steps in the process of completing specific web tasks, meaning that regardless of the path taken to accomplish a task, these steps are required. These may involve navigation to certain webpages or the performance of specific actions on web pages, such as filling out forms or clicking buttons. This design philosophy not only reflects the dynamic nature of the web environment but also captures the diversity of paths present in real-world web pages.

As illustrated in Figure 2, consider the task of "Find top-rated upcoming adventure movies on Rotten Tomatoes" as an example. Users might start directly at the Rotten Tomatoes homepage or use a search engine to navigate straight to the "New Movies Coming Soon" page of the Rotten Tomatoes. Moreover, when filtering the movies, users might choose to first apply a filter for the "adventure" genre and then sort by popularity, or alternatively, sort by popularity before applying the genre filter. Despite the availability of different paths to achieve the goal, entering the specific page and performing the genre and popularity sorting are essential steps in accomplishing the task. Therefore, these three steps are identified as "key nodes".

In the dynamic and noisy real-world web environment, identifying these key nodes is challenging due to the potential changes in page content and UI updates, which could render element selector paths obsolete. Therefore, we preferred to use URL state as identifiers for key nodes rather than element interaction, which enhanced the Benchmark's robustness against layout changes. Only element class methods are considered for key nodes that cannot be represented by URLs. The detailed judgment method is described in Appendix C.3. By defining key nodes, WebCanvas is able to dynamically assess the execution capabilities of web agents in real-world web environments, offering a practical and flexible evaluation method for the development of web agents.

## 2.3. Evaluation Metrics

The evaluation metrics of WebCanvas comprised of two main components: step score and task score. The step score evaluates the agent's performance with regard to each key node, defining three types of evaluation targets along with three evaluation functions at each step. The task score includes two functions to assess the task's completeness and overall execution efficiency.

### 2.3.1. STEP SCORE

Inspired by previous works (Zhou et al., 2023; Koh et al., 2024), we introduced three evaluation targets in calculating step score, allowing us to examine from different aspects: **(1) URL:** The webpage's URL, used for page location and parameter retrieval. **(2) Element Path:** Selector to pinpoint element location. **(3) Element Value:** The text content of the target element.

We designed three evaluation functions for these targets: **(1) Exact Match:** This function requires the agent's output to exactly match the reference answer, suitable for scenarios requiring precise matches, such as specific URL parameters or form fields like names. **(2) Include Match:** This function scores the agent's output as long as it contains reference answer, suitable for keyword matching scenarios. **(3) Semantic Match:** Utilizing LLM for semantic matching scoring, this function is appropriate for complex tasks requiring content understanding, like identifying product information.

Each key node is bonded with a specific evaluation target and evaluation function. One step score is awarded when the agent successfully reaches a key node and passes the associated evaluation function verification. Table 5 shows a list of possible evaluation functions for reference. To facilitate the presentation of experimental results, the "Completion Rate" will be used to represent the proportional scoring of Step Scores.

### 2.3.2. TASK SCORE

**Task Finish Score** Task Finish Score is awarded based exclusively on the agent's success in completing all the designated key nodes within the task. To facilitate the presentation of experimental results, the Task Finish Score will be represented by the "Task Success Rate".

**Efficiency Score** Efficiency Score(ES) is devised to evaluate the resource utilization effectiveness during task execution, which is calculated based on the average number of steps required for the agent to achieve each unit of the step score, thereby encouraging maximal efficiency with minimal resource expenditure.

## 2.4. Data

### 2.4.1. DATASET: MIND2WEB-LIVE

To develop a real-world online benchmark for web agents, we introduce Mind2Web-Live, which is derived from tasks present in the Mind2Web dataset. We employed WebCanvas

framework as a guidance for the sampling and re-annotation of these tasks to ensure their adaptability. Consequently, we selectively excluded all tasks that contained time-sensitive descriptions, such as those involving specific dates or times. From the training set, we randomly sample 601 tasks, along with all 179 tasks from the cross-task subset of the test set. These tasks are then re-annotated in the real-world online environment.

The annotation process presented multiple challenges. Notably, due to updates in website content and operational changes, we discovered 96 tasks that were no longer applicable and subsequently removed them from the dataset. Additionally, 142 tasks were discarded due to ambiguous task definitions and the difficulty in clearly defining key nodes. To enhance the clarity and reliability of task execution, we revised the descriptions for 51 tasks.

After a rigorous annotation and review process, which is described in Appendix B due to length limit, 542 high-quality tasks were established for the Mind2Web-Live dataset, including 438 of the training set and 104 of the test set. As shown in Figure 1, Mind2Web-Live encompasses 2439 key nodes and 4550 detailed annotation steps. The tasks in the dataset cover a wide range of webpage types and operations, designed to comprehensively evaluate the performance of web agents in a dynamic and variable online environment. The distribution of the Evaluation Function within the dataset is illustrated in subsection C.2.

Table 1: Data distribution

| Statistic | Number |
|---|---|
| Total selected tasks | 780 |
| - Expired Tasks | 96 |
| - Unable to annotate | 142 |
| **- Mind2Web-Live** | **542** |
|   - training set | 438 |
|   - testing set | 104 |
| Annotate steps | 4550 |
| Avg. steps | 8.39 / task |
| Eval functions | 2439 |
| Avg. Eval functions | 4.5 / task |

### 2.4.2. DATASET MAINTENANCE

We have paid special attention to the dynamic nature of the benchmark to adapt to the constantly changing web environment. We recognize that updates and changes to website content, such as UI updates, database changes, or website close-down, are inevitable as time progresses. Such changes may lead to the obsolescence of previously defined tasks or key nodes.

We thus implemented a regular data maintenance schedule. During data collection process, in addition to key nodes annotation, we recorded detailed information about workflow execution, including action types, selector paths, element value, and element coordinates at each step. We managed to stably playback these stored action workflows by an element matching strategy in our replaySDK[1], and report any invalidity in the workflows or the evaluation functions. We periodically reassess key nodes by the above methods and a human check to ensure that each task reflects the current web environment, as illustrated in Appendix H. An example of regular testing report is shown in Appendix D.

## 3. Experiment

Inspired by previous work (Yao et al., 2023; Zhou et al., 2023; Zheng et al., 2024), we introduce a universal agent framework, as illustrated in subsection G.2, which includes four key modules: Planning, Observation, Memory and Reward. This framework is engineered to perform complex tasks within real-world online web environments.[2] Experimental settings are detailed in Appendix I.

### 3.1. Agent Framework

**Planning** Integrates past states' history, current observations, and task queries to plan future actions and determine operational values based on the ReAct (Yao et al., 2023) reasoning framework. It can be formally expressed as: $\textbf{Planning}(\mathbf{h_1^t}, \mathbf{o_t}, \mathbf{i}) \longrightarrow (\mathbf{z_t}, \mathbf{a_t})$, where $\mathbf{h_1^t}$ represents history information until time $\mathbf{t}$, $\mathbf{o_t}$ is the observation at time $\mathbf{t}$, $\mathbf{i}$ is the task instruction, while the outputs $\mathbf{z_t}$ and $\mathbf{a_t}$ are the thought and action at time $\mathbf{t}$ respectively.

**Observation** Processes the current webpage's source code and screenshots, producing an accessibility tree (Zhou et al., 2023) and visual observations as $\mathbf{o_t}$. Observation settings are detailed in Appendix I.

**Memory** Responsible for storing the task description and tracking the agent's operational history, including thoughts and actions history across states. It can be formally expressed as $\mathbf{h_1^t} = (\mathbf{z_1^t}, \mathbf{a_1^t}, \mathbf{r_1^t})$ within the framework, where $\mathbf{r_1^t}$ denotes the history of reward signal if presents.

**Reward** Utilizes a self-reflection structure (Shinn et al., 2024), providing a series of reward signal, including a verbal reflection and signal on whether the task is completed. This can be formalized as $\textbf{Reward}(\mathbf{h_1^t}, \mathbf{i}, \mathbf{o_{t+1}}) \rightarrow \mathbf{r_t}$.

---

[1] While the full codebase is not currently available, we intend to open source the code upon completion of further refinement and documentation

[2] https://anonymous.4open.science/r/WebCanvas_Agent-A2C3/README.md

Table 2: Performance of different models without reward module on Mind2Web-Live test set. "Task Success Rate(0)" and "Task Success Rate(1)" denote the Task Success Rates with zero tolerance and tolerance for error at one key node, respectively. As for the model, we experiment with gpt-3.5-turbo-0125(GPT-3.5), gpt-4-0125-preview(GPT-4).

| Model | Completion Rate | Task Success testRate(0) | Task Success Rate(1) | Efficiency Score |
|---|---|---|---|---|
| GPT-3.5 | 40.2% | 16.5% | 32.0% | 3.03 |
| GPT-4 | **48.8%** | **23.1%** | **40.3%** | **2.47** |
| Claude-3-Opus | 32.1% | 16.3% | 24.9% | 4.23 |
| Gemini-Pro | 35.3% | 13.4% | 31.6% | 4.69 |
| DeepSeek-V2 | 41.2% | 18.2% | 32.6% | 4.44 |
| Mixtral-8x7B | 3.37% | 0.00% | 1.19% | 61.8 |
| Mixtral-8x22B | 37.2% | 17.3% | 28.8% | 4.80 |

## 3.2. Main Results

In this study, we extensively explore the impact of various combinations of planning models and reward models on the performance of web agents in online web tasks. We employ reward module to determine whether a process has been completed. In experiments without a reward module, we set a maximum execution step length of 1.2 times the annotated task length. Table 2 indicates that while GPT-4 outperforms other models in both effectiveness and efficiency in live environment, overall performance across all models remains considerable room for future enhancements. In Appendix, we conduct a detailed qualitative and quantitative analysis, in Appendix K and Appendix J respectively, to identify the main obstacles faced by web agents in live environments, and to analyse factors that influence web agent performance and evaluation.

## 4. Related Works

**Agent Benchmarks**   Early researches (Shi et al., 2017) (Liu et al., 2018) provided relatively simple simulations and assessment methods for web navigation tasks. However, with the rise of Large Language Models, these methods have become inadequate for assessing agents' capability. Recent studies have chosen to construct realistic simulated environments (Yao et al., 2022) (Zhou et al., 2023) (Koh et al., 2024) (Drouin et al., 2024), use offline saved datasets (Deng et al., 2024) (Lù et al., 2024), or select relatively stable answers to assess the capabilities of web agents (Mialon et al., 2023). In terms of dynamic evaluation methods, many studies (Kiela et al., 2021) (Ma et al., 2021) (Jain et al., 2024) have proposed their own solutions. Moreover, beyond network platforms, several initiatives have also been undertaken on other platforms such as Android mobile devices, operating systems, and databases (Rawles et al., 2024; Liu et al., 2023; Xie et al., 2024). As shown in Table 3, WebCanvas aims to more comprehensively test agents' capability in the real world through key nodes and corresponding evaluation functions.

**Agent Frameworks**   In the area of reasoning frameworks, several studies have achieved notable success in logical reasoning challenges (Wei et al., 2022; Yao et al., 2024; 2023; Shinn et al., 2024; Sumers et al., 2024). Regarding web agent reasoning frameworks, many researches has been conducted to enhance the capabilities of web agents (Nakano et al., 2021; Gur et al., 2023; Gür et al., 2023; Kim et al., 2024; Lo et al., 2023; Lai et al., 2024). Some studies have introduced multimodal modules that integrate visual and semantic information, thereby enhancing the capabilities of agents on web platforms (Zheng et al., 2024; Furuta et al., 2023; He et al., 2024).

## 5. Conclusion

In this work, we have pioneered the development of framework for online evaluation of web agents, and investigated the challenges associated with online evaluation and the difficulties faced by current web agent reasoning frameworks in online inference. Simultaneously, we have constructed a community-driven platform that empowers web agent researchers and developers to build datasets and evaluate their web agent frameworks and models in an online environment while collecting feedback on dataset design, data annotation quality, and data validity throughout the process. We strongly encourage further work on online datasets, web agents, and evaluation function designs. By fostering a collaborative and iterative value to dataset creation and evaluation, we eagerly anticipate the continued growth of advancement of autonomous intelligence.

## Impact Statement

**Ethical Impact:** The technologies developed in this research could potentially enhance the capabilities of web crawlers, thereby exacerbating issues related to personal privacy and data security. To mitigate these potential risks, we specifically avoid using websites that involve sensitive information in designing our benchmark. We emphasize

using our technology in compliance with website usage agreements and data protection regulations. Furthermore, our benchmark does not include any processes that require user login or involve personal information and avoids any irreversible actions. The selection of websites and processes is entirely transparent. Additionally, the widespread adoption of web automation technology could alter the nature of human work, substituting certain types of employment, thus causing structural changes in the labor market.

**Societal Impact:** On the positive side, this research could improve the efficiency of various online services, such as online customer support and data retrieval, potentially enhancing overall economic efficiency and user experience. However, this may also exacerbate the digital divide, as technological advancements may initially benefit technically advanced organizations and individuals, widening the gap with other societal groups.

We encourage community members and policymakers to pay attention to these potential issues and adopt appropriate regulatory measures when using our technology. Additionally, our research provides open access to data and models, promoting transparent and responsible scientific practices to foster healthy development in this field.

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

# A. Case study of previous benchmarks

See Table 3.

Table 3: Case study of previous benchmarks

| Benchmark | Real-world Intents | Dynamic Environment | Keep Updated | Intermediate Env. State | Easy to Scale | Disk Usage |
|---|---|---|---|---|---|---|
| MiniWoB++ | ✗ | ✓ | ✓ | ✗ | ✗ | $< 1$GB |
| WebShop | ✗ | ✓ | ✗ | ✗ | ✗ | $\sim 10$GB |
| Mind2Web | ✓ | ✗ | ✗ | ✗ | ✗ | $\sim 10$GB |
| WebArena | ✓ | ✓ | ✗ | ✗ | ✓ | $> 100$GB |
| VWebArena | ✓ | ✓ | ✗ | ✗ | ✓ | $> 100$GB |
| GAIA | ✓ | ✓ | ✗ | ✗ | ✓ | $< 1$GB |
| WEBLINX | ✓ | ✗ | ✗ | ✓ | ✗ | $< 1$GB |
| OmniACT | ✓ | ✗ | ✗ | ✓ | ✗ | $< 1$GB |
| **WebCanvas** | ✓ | ✓ | ✓ | ✓ | ✓ | $< 1$GB |

# B. Data Collection Details

## B.1. Recording process

In the construction of Mind2Web-Live, the quality and reliability of the data are paramount. To this end, we have employed the iMean Builder plugin[3], an efficient tool for recording browser operations. This tool precisely captures browser interaction from the users, covering a wide range of activities such as clicks and input actions. The recorded details include the type of operation, execution parameters, target element's selector path, element content, and its coordinates on the webpage. Moreover, iMean Builder accompany each step with a webpage screenshot, not only facilitating process replication but also providing a visual reference for workflow validation and review. This approach enables us to comprehensively record all the steps required to complete specific tasks, forming the foundation of Mind2Web-Live. Upon completion of the data recording, we meticulously annotated the key nodes of each process along with their corresponding Evaluation Functions.

## B.2. Annotation process

In our study, the annotation process plays a pivotal role in ensuring data quality and task validity. To ensure the accuracy and consistency of data annotations, we assembled an annotation team comprised of several authors of this paper and five senior undergraduate students majoring in Computer Science. Not only do the members of the annotation team possess a solid background in Computer Science, but they also received specialized training to ensure consistency in their understanding and identification abilities in annotating key nodes.

During the annotation phase, we employed a comprehensive reward mechanism. Each annotator was compensated based on the number of tasks they completed, with additional bonuses awarded for high-quality annotations to encourage precise and consistent results. This combined reward system not only bolstered work enthusiasm but also enhanced the overall quality of the annotation work, laying a solid foundation for the construction of an efficient web agent benchmark.

To guarantee the quality of annotations, we instituted a variety of strategies. Each task was annotated independently by one annotator, followed by individual reviews by two other members to verify the accuracy of the key nodes. Throughout the annotation process, we regularly organized discussion sessions for the annotation team to share their experiences and challenges encountered, thereby improving the overall efficiency and quality of the team's annotations.

---

[3]https://builder.imean.ai/

Table 4: Example Annotations of the Evaluation Functions

| State | Title | Annotation Details |
|---|---|---|
| | Locate a large store in Washington that has kids' and maternity products in uniqlo | Evaluation Function: `Element value semantic match` 

 Instructions: `Decide Whether is searching for Washington D.C.` |
| | Find parking in California city for Limos which also offers free wi-fi in yelp | Evaluation Function: `URL include match` 

 Param: `attrs` 
 Value: `WiFi.free` |
| | Find Dota 2 game and add all DLC to cart in steam | Evaluation Function: `Element path exact match` 

 Selector: `//*[@id="dlc_purchase_action"]/div[2]/a/span` |

# C. Details of evaluation functions

## C.1. Available evaluation functions

See Table 5

## C.2. Evaluation Function distribution

See Figure 3

Table 5: Evaluation functions

| Evaluate target | Exact match | Include match | Semantic match |
|---|---|---|---|
| URL | ✓ | ✓ | ✓ |
| Element path | ✓ | ✗ | ✗ |
| Element value | ✓ | ✓ | ✓ |

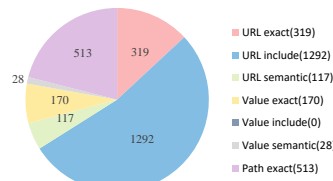

- URL exact(319)
- URL include(1292)
- URL semantic(117)
- Value exact(170)
- Value include(0)
- Value semantic(28)
- Path exact(513)

Figure 3: Evaluation Function distribution.

## C.3. How to define evaluation functions

**For input operations on the page** First, determine whether it is a necessary condition for task completion. If it is a necessary condition, then judge whether the execution result can be reflected by the change of the URL. If so, simply take the state after execution as the key node and select the evaluation function as URL exactly/included/semantic match.

If it cannot be reflected by changes in the URL, it needs to be defined as a key node based on click or input operations. Select element path exactly match or element value exactly/included/semantic match for input operations (to determine whether the content of the input element matches).

**For click operations on the page** Firstly, determine whether it is a necessary condition for completing the task. If it is a necessary condition, then judge whether the execution result can be reflected by the change of the URL. If so, simply take the state after execution as the key node and select the match rule as URL exactly/included/semantic match.

If it cannot be reflected by the change of URL, each click operation should be defined as a key node, and the match can be selected as element element path exactly match or element value match.

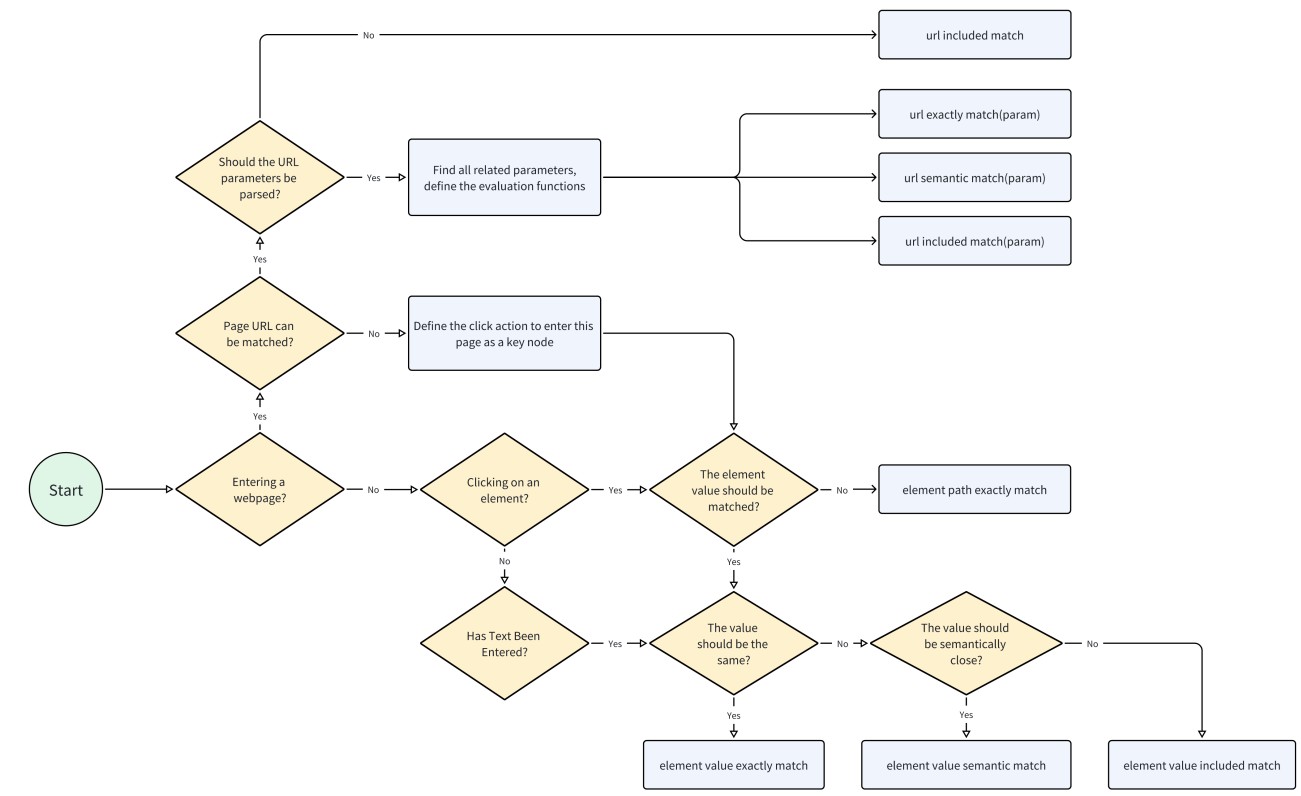

Figure 4: Guide to defining an evaluation function

# D. Data Validity Test Report

See Figure 5.

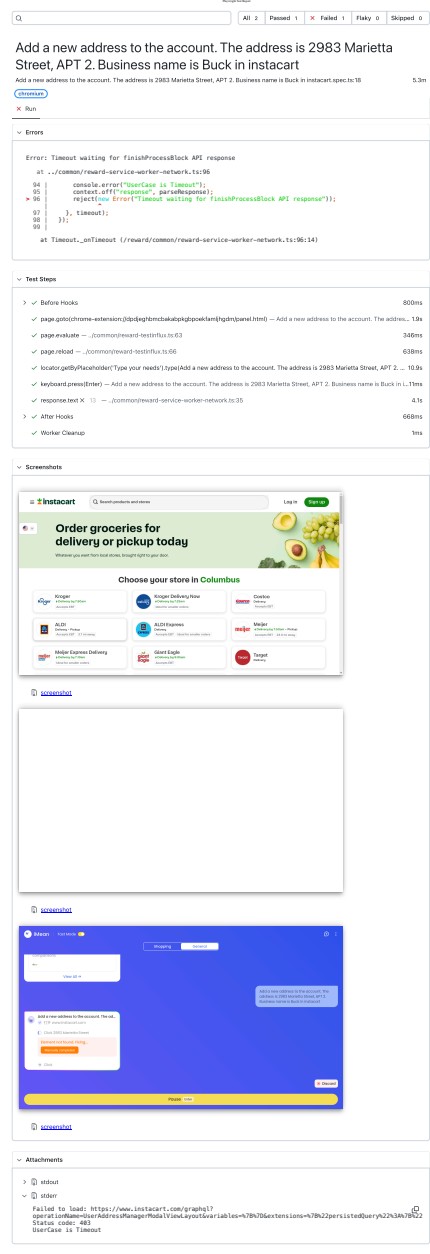

Figure 5: Data Validity Test Report

# E. Additional Evaluation Metrics

**Human Alignment Score**  The Human Alignment Score(HAS) assesses how well an agent's workflow aligns with human behavior. It's crucial for agents not just to be efficient, but to operate in ways that resemble human actions. The evaluation of this aspect is conducted by contrasting the agent's task completion signal with the ground truth annotations provided by humans, to gauge the level of consistency. An agent that accurately issues a completion signal upon task completion is deemed to exhibit a high degree of alignment with human behavior, thus earning a full score of one point. Conversely, a delay in issuing the completion signal upon task completion results in a deduction of 0.05 points from the full score as a penalty for decision latency. In instances where an agent stops its operation before accomplishing all the task objectives, the score is determined by the ratio of the step score attained to the maximum step score achievable for that task. Furthermore, if a task is not fully completed and the system forcibly terminates the process due to reaching the maximum step limit, the score awarded is 0.8 times the proportion of the step score attained. The specific algorithm is shown in the formula, where $P$ represents achieved step scores, $P_{max}$ denotes the max step scores of the task.

$$HAS = \begin{cases} 1 & \text{if task is completed with completion signal} \\ 0.95 & \text{if task is completed without completion signal} \\ \frac{P}{P_{max}} & \text{if task is incomplete but completion signal} \\ 0.8 \times \frac{P}{P_{max}} & \text{if task is incomplete and is terminated} \end{cases} \quad (1)$$

# F. Comparison of the Mind2Web-Live and Mind2Web Datasets

See Table 6.

Table 6: Comparison of the Mind2Web-Live and Mind2Web Datasets. "Ele." indicates "Element", "Op." indicates "Option" and "SR" indicates "success rate".

| Attributes | Mind2Web-Live | Mind2Web |
|---|---|---|
| Dataset Size | 438 | 2350 |
| Evaluation Environment | Real-world Online | Offline |
| Evaluation State | Key Nodes | Each Step |
| Target Element | Element, URL | Element, Option |
| Evaluation Metrics | Step Score & Task Score | Step(Ele., Op.) SR & Task SR |
| Avg. Steps | 8.51 / task | 7.3 / task |

# G. Agent Framework

## G.1. Action Space

See Table 7.

Table 7: Action Space

| Action | Operation value |
|---|---|
| Goto | Value |
| Google Search | Value |
| Click | Target id |
| Hover | Target id |
| Fill Form | Target id, value |
| Fill Search | Target id, value |
| Switch Tab | Target id |
| Go Back | / |

### G.2. Diagram of Agent Framework

See Figure 6.

## H. Diagram of Data Maintenance

See Figure 7.

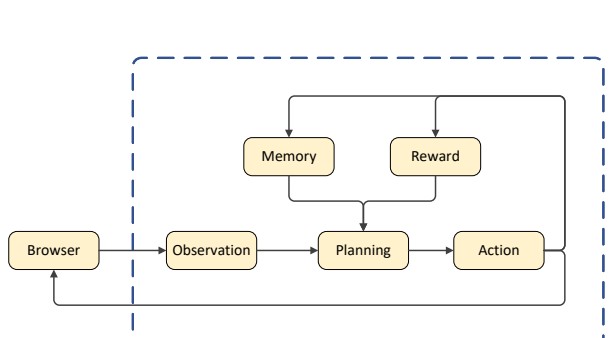

Figure 6: Agent Framework

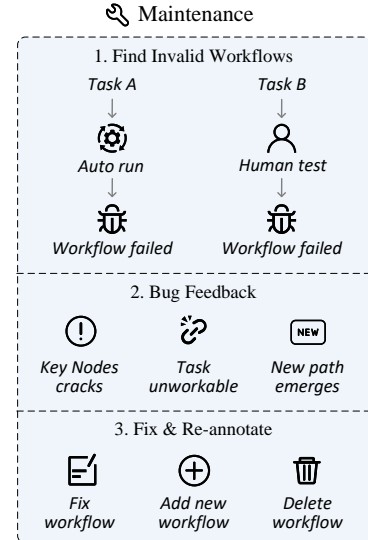

Figure 7: Illustration of how maintenance system works.

## I. Experimental Settings

### I.1. Observation Space

**Accessibility Tree**    We employ an accessibility tree-based approach to extract the fundamental textual feature representation from the web environment. The accessibility tree serves as an abstract representation of the structure of a web page, detailing the characteristics of each element within the page. However, the accessibility tree contains a significant amount of redundant information, necessitating the use of a stringent set of filtering criteria to select interactive elements. These filtering criteria include the element's tag, visibility, usability, as well as textual or image content. Concurrently with the construction of the accessibility tree, we annotate each filtered interactive element, providing information such as element ID, tag, and content. For example, ([1] input 'search', etc.). This annotation method facilitates the precise generation of corresponding CSS selector paths during subsequent LLM prediction and execution phases, thereby accurately locating the required elements.

**Screenshot**    We capture screenshots of the current web page to obtain its visual representation and provide this visual context to visual language models, such as GPT-4V. This input method mimics human visual perception, allowing the model to gather the most comprehensive information from the web page. Compared to relying solely on the accessibility tree, using screenshots enhances the ability to identify the layout, appearance, and positioning of web elements more effectively. Additionally, it captures interactive elements and other crucial page information that the accessibility tree might miss. To balance inference costs and recognition effectiveness, the original resolution of the screenshots is set to 1080 x 720, though users can define the screenshot resolution according to their specific needs in practical applications.

Table 8: Experiments on implementation of reward module. "(+)" indicates the inclusion of a reward module with golden reference. Model notation follows Table 2, except for gpt-4-vision-preview(GPT-4V). Human Alignment score represents agents' alignment with human decision on task completion, while the larger indicates better alignment, detailed in Appendix E.

| Planning Model | Reward Model | Completion Rate | Task Success Rate | Efficiency Score | Human Alignment |
|---|---|---|---|---|---|
| GPT-3.5 | / | 34.2% | 13.0% | 5.29 | / |
| GPT-4 | / | 48.3% | **16.7%** | 3.77 | / |
| GPT-4 | GPT-3.5 | 42.9% | 14.6% | 3.28 | 0.440 |
| GPT-4 | GPT-4 | 41.7% | 12.3% | 3.10 | 0.426 |
| GPT-3.5 | GPT-4 | 36.6% | 10.8% | 3.73 | 0.385 |
| GPT-4 | GPT-4V | 42.4% | 8.3% | 3.42 | 0.419 |
| GPT-3.5 | GPT-4(+) | 43.6% | 13.8% | 3.28 | 0.452 |
| GPT-4 | GPT-4(+) | **52.3%** | 12.5% | 3.27 | **0.506** |
| GPT-4 | GPT-4V(+) | 51.3% | 12.5% | **2.71** | 0.502 |

## J. Quantitative Analysis of Experiments

### J.1. Planning with Golden Reward Reference

We contemplated the impact of the quality of the reward signal on the web agent performance, raising a natural question - *Can high-quality reward signals lead to better agent performance?* In our study, we introduced a reward module with golden reference. The experimental results on Mind2Web-Live, which confirm our hypothesis, are detailed in Table 8.

From the original data, we extracted post-action URLs, action types, CSS selector paths, and key nodes functions as metadata for our golden reference synthesis. We then employed a carefully designed prompt (available in Appendix O, using GPT-4 to generate a structured linguistic guidance for task progress estimation for each task. This guidance includes the overall goal of the current task and task completion criteria (specifically highlighting all key nodes that must be met for the task to be considered fully completed). We then integrate the content of the current task's golden reference with the original design of history and current observation for reward reasoning.

#### J.1.1. RESULTS

We found that in the complex web environment, the capability of language models to utilize self-generated feedback is limited. The integration of a reward module does not enhance agent performance and may even lead to a decline in Task Success Rate and Task Completion Rate. This is often due to the models prematurely concluding the tasks, whether with textual or visual observations. This finding aligns with findings in (Shinn et al., 2024) about the effect of self-reflection modules in web agent tasks.

Regardless of the planning model used, whether GPT-3.5 or GPT-4, the performance of model inference improves with the integration of a reward module with golden reference, particularly in terms of Task Completion Rate and Task Efficiency Score. This enhancement is primarily due to the reward model's better alignment with human understanding of task completion.

### J.2. Task Complexity & Task Difficulty

In this section, we investigate the factors influencing the agent performance and their correlation with agent performance. Our analysis primarily focuses on the relationship between the complexity of a task, as quantified by the number of steps annotated(step count) and the number of key nodes needed to complete the task, and the performance metrics of agent completion and success rates.

Figure 8A and Figure 8B highlights a discernible trend: as the step count and the number of key nodes increases, there is a notable decline in both the task completion rate and the success rate, suggesting that tasks involving more steps and more key nodes are inherently more challenging.

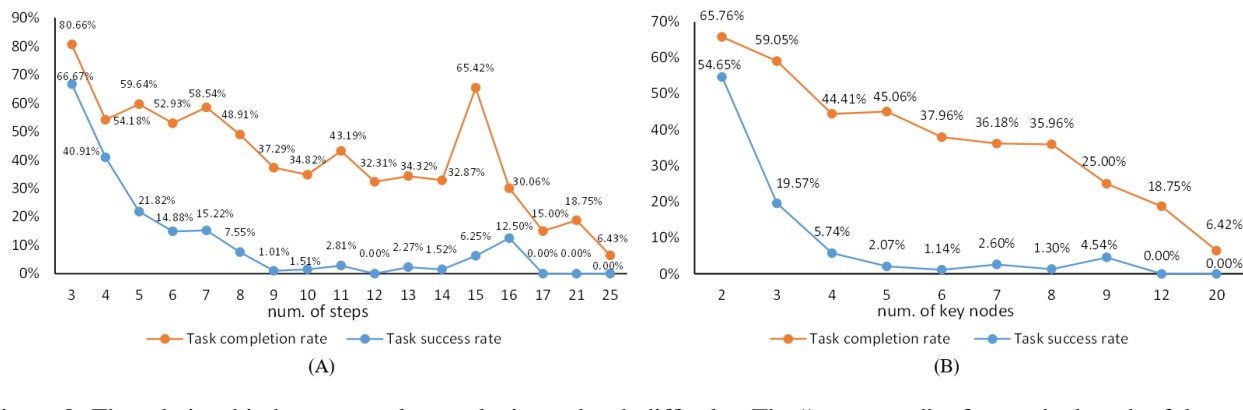

Figure 8: The relationship between task complexity and task difficulty. The "step count" refers to the length of the action sequence in the annotated data, which, along with the number of key nodes, serves as a reference for task complexity.

### J.3. Discrepancy between Offline and Online Evaluation

The settings of evaluation on offline datasets that reflect real-world intents, such as Mind2Web (Deng et al., 2024), are inherently different from WebCanvas framework. Nevertheless, we managed to compare the experimental results between offline and online testing. During online inference, we attempted to reproduce the setting of the MindAct model, which was trained and evaluated on the offline dataset, as proposed in the Mind2Web paper. It is important to note that the evaluation metrics used in offline evaluation differ from those proposed in our online evaluation framework. The Step Success Rate in offline testing assesses the accuracy of single-step action prediction, and for the entire task dimension, a positive reward is given only when all single-step actions are correctly predicted (which is not the case in online evaluation, as we evaluate the intermediate state, not the referenced action). As shown in Table 9, we have two main findings:

1. The model trained on the Mind2Web training set does not generalize well to the online environment one year later. The comparative relationship between the results of MindAct-Large (Deng et al., 2024), GPT-3.5, and GPT-4 is the opposite of that in offline testing.

2. The metrics used in offline testing only evaluate the accuracy of action prediction and do not consider the complexity of the decision space in the real-world environment. Consequently, the Task Success Rate of GPT-3.5 and GPT-4 in offline testing is inconsistent with the results in online testing.

Table 9: Comparison of web agent performance in online and offline evaluations. We sampled 40 instances from the Mind2Web test set and annotated them according to the WebCanvas framework to define key nodes. These were then tested in both online and offline settings. 'Task SR(0)' and 'Task SR(1)' denote the Task Success Rates with zero tolerance and tolerance for error at one step (or key node), respectively.

| Model | Offline | | | Online | | |
|---|---|---|---|---|---|---|
| | Step SR(%) | Task SR(0)(%) | Task SR(1)(%) | Completion Rate(%) | Task SR(0)(%) | Task SR(1)(%) |
| MindAct | 44.3 | 10.0 | 25.0 | 25.5 | 7.50 | 12.5 |
| GPT-3.5 | 15.5 | 2.50 | 7.50 | 35.4 | 10.0 | 17.5 |
| GPT-4 | 28.4 | 5.00 | 22.5 | 41.1 | 10.0 | 25.0 |

## K. Qualitative Analysis of Experiments

In this section, we conducted a qualitative analysis of error cases in our experimental results. Typical errors include: local optima, premature termination of tasks, and information loss during inference.

### K.1. Local Optima

In our online environment experiments, the result data shows that a task may involve multiple constraints or requirements, or it may require navigating through multiple web states within a specific domain. Web pages often contain numerous clickable links, and frequently feature interactable elements with similar or even identical names. This complexity demands that web agent faces an almost exponential growth in the decision space as the number of steps increases, while precisely selecting the web element that can complete the task and appropriately advancing the fulfillment of multiple constraints. Specifically, our web agent tends to operate on an unidirectional forward path (described in K.3), making it difficult to revert to an intermediate state within a limited number of steps. Furthermore, due to a lack of prior knowledge about the web domain associated with current task and confusion caused by similar elements, the planning module's local decision-making for the current web state is not always accurate, and it also lacks proactive thinking to explore alternative paths. This is one of the main reasons for the low task success rate. As shown in Table 10, in the task "Check the rating and user reviews for the game 'Deathloop' on IGN", the web agent ended up at the review article page for 'Deathloop' on IGN due to incorrect path selection from the Google search results, rather than the expected page for ratings and user reviews.

### K.2. Premature Termination of Tasks

In the experiments, we also discovered that the web agent sometimes only partially completes tasks. This typically indicates that web agent sometimes prematurely judges itself as having finished the task. The reasons for premature termination are varied. For instance, the agent might hallucinate during inference (such as simplifying a task of reaching a page and filling out content to just reaching the page), leading it to self-judge the task as complete after only finishing intermediate steps and not continuing further. In other instances, it may have the right thought process in earlier steps, but fails to deliver the correct action input or effectively execute the action on the page, yet in subsequent steps, it "reads" this thought and mistakenly believes the action has been executed. Lastly, when it is difficult to continue along the current path, the agent might lower its standards for task completion and erroneously judge the task as complete, thus terminating the task prematurely. As shown in Table 10, in the task "Track the status of a rebate for Monty Lue, house number 4847, zip code 10019 in Menards", the web agent reached the "Track Your Rebate" page but did not continue to complete the form, instead prematurely deciding the task was complete and ending the task.

Table 10: Case Study of Failure Trajectories

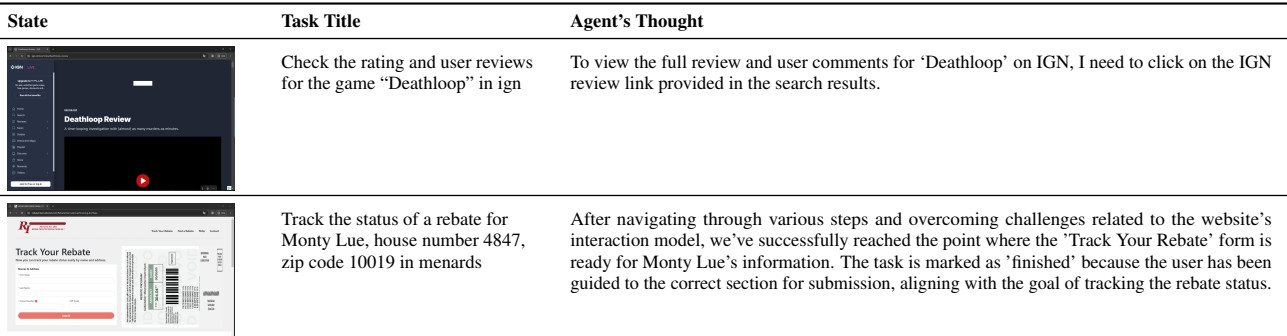

| State | Task Title | Agent's Thought |
|---|---|---|
| | Check the rating and user reviews for the game "Deathloop" in ign | To view the full review and user comments for 'Deathloop' on IGN, I need to click on the IGN review link provided in the search results. |
| | Track the status of a rebate for Monty Lue, house number 4847, zip code 10019 in menards | After navigating through various steps and overcoming challenges related to the website's interaction model, we've successfully reached the point where the 'Track Your Rebate' form is ready for Monty Lue's information. The task is marked as 'finished' because the user has been guided to the correct section for submission, aligning with the goal of tracking the rebate status. |

### K.3. Information Loss during Inference

**Element Mapping** The relationships between web elements are varied and complex. Often, the essential information of an element is not contained within the element itself but is instead found within its child elements, parent, or even sibling elements. For instance, a button tag might not always contain useful attributes; sometimes, they are empty or irrelevant. Based on our understanding of the DOM tree on the web, we map information from specific elements (like span) to interactive elements such as buttons. Due to the diversity of these mapping relationships, our framework currently only

considers mapping valuable information from certain special elements to their parent elements, recursively iterating until an interactive element is identified, as shown in Figure 9 in Appendix L. If this recursive search fails to find an interactive element or reaches the recursion limit, the element is discarded, as illustrated in Figure 10 in Appendix L. Given the complexity of webpage elements, our initial implementations focus predominantly on parent-child mapping relationships. Future work will delve deeper into inter-element mappings to ensure the accuracy and correctness of element mappings.

**Unidirectional Execution**   Actions planned web agents depend on the attributes fetched from the network environment. However, our analysis of the execution process reveals that these planned actions are often tied to specific web page elements. When these actions are executed, such as clicking a link, the page may redirect to unpredictable pages, leading to several issues during the execution. Our analysis has identified two primary types of problems:

1. Pages may load to irrelevant sites due to network issues, access restrictions, or login requirements, resulting in navigation to pages like blank screens or CAPTCHA verification, exemplified in Figure 11B.

2. Although an action may lead to a reasonable page, extracting relevant information from this page can be challenging. As shown in Figure 11A, without additional vision or OCR processing, the action sequence stalls.

The limitations of browser automation tools currently prevent the complete restoration of a web page to its state before action execution. Meanwhile, memory management of web agents also could not eliminate the effect of past incorrect trajectories. Therefore, task completion often falls into a loop state. Despite efforts to selectively remove ineffective or infeasible actions from the historical trajectory, these problems persist, highlighting the challenges of autonomous web interaction.

## L. Additional Examples on Case Study

See Figure 9, Figure 10, Figure 11

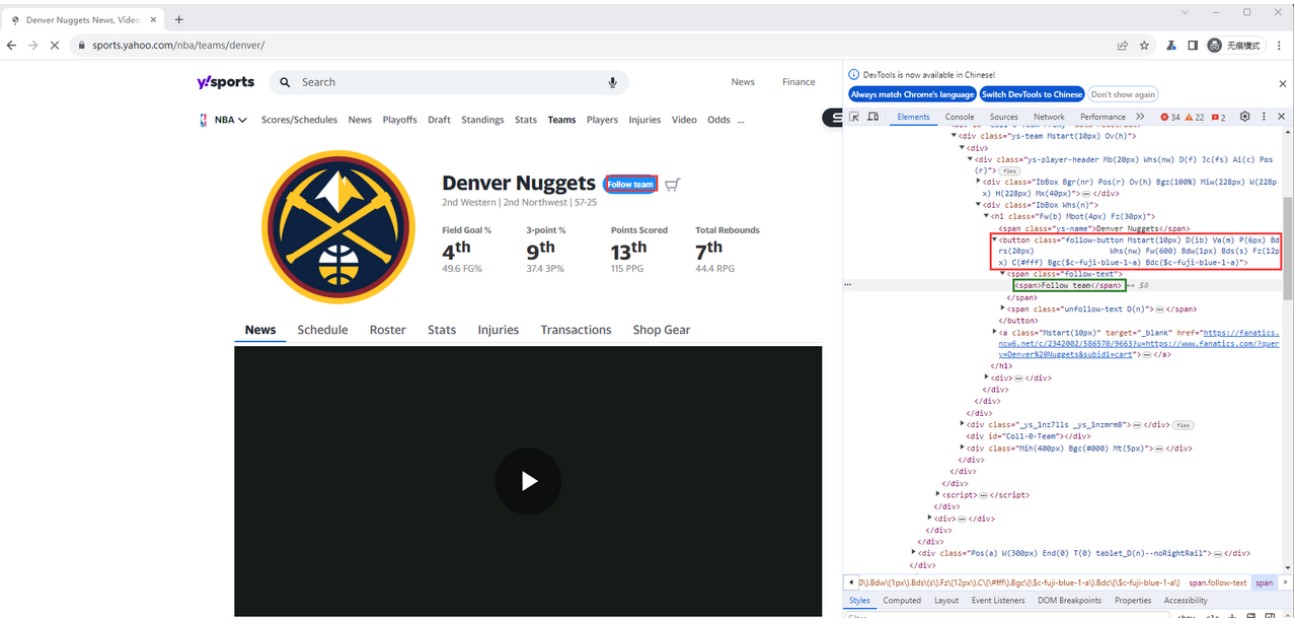

Figure 9: Example on parent-child element mapping strategy(1).

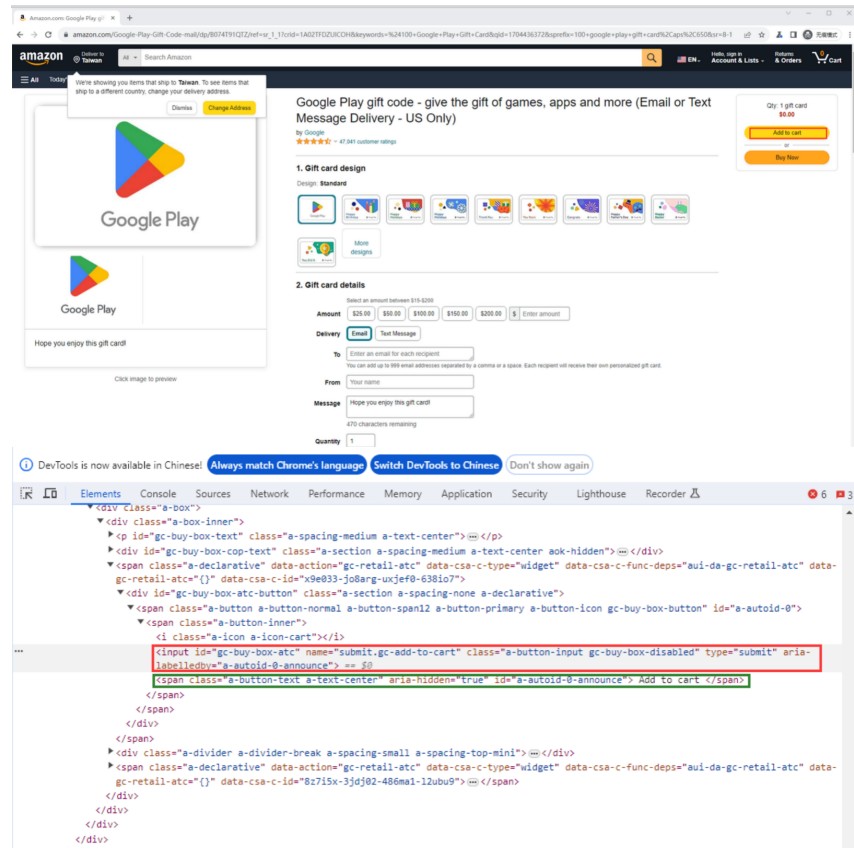

Figure 10: Example on failure case of parent-child element mapping strategy(2).

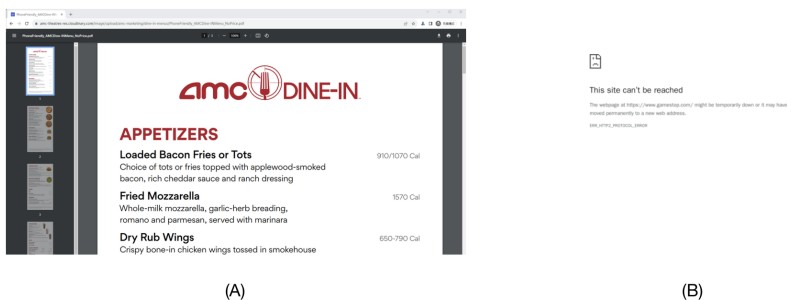

(A)                                                                          (B)

Figure 11: Examples on unidirectional failure case.

# M. Examples of More Annotated Samples

See Figure 12, Figure 13.

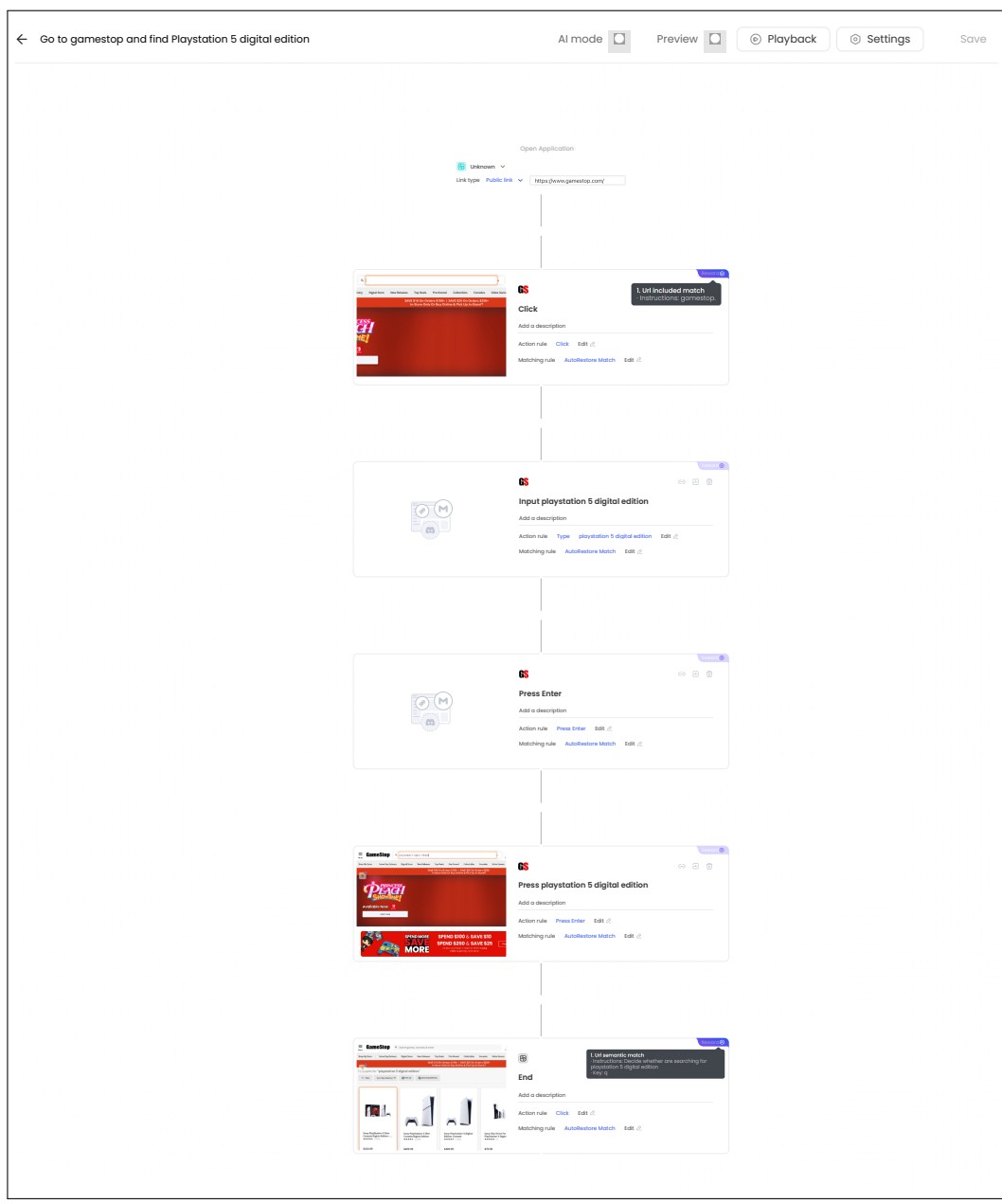

Figure 12: Example on the Annotated Interface and Evaluation Function for the Task "Go to GameStop and Find PlayStation 5 Digital Edition"

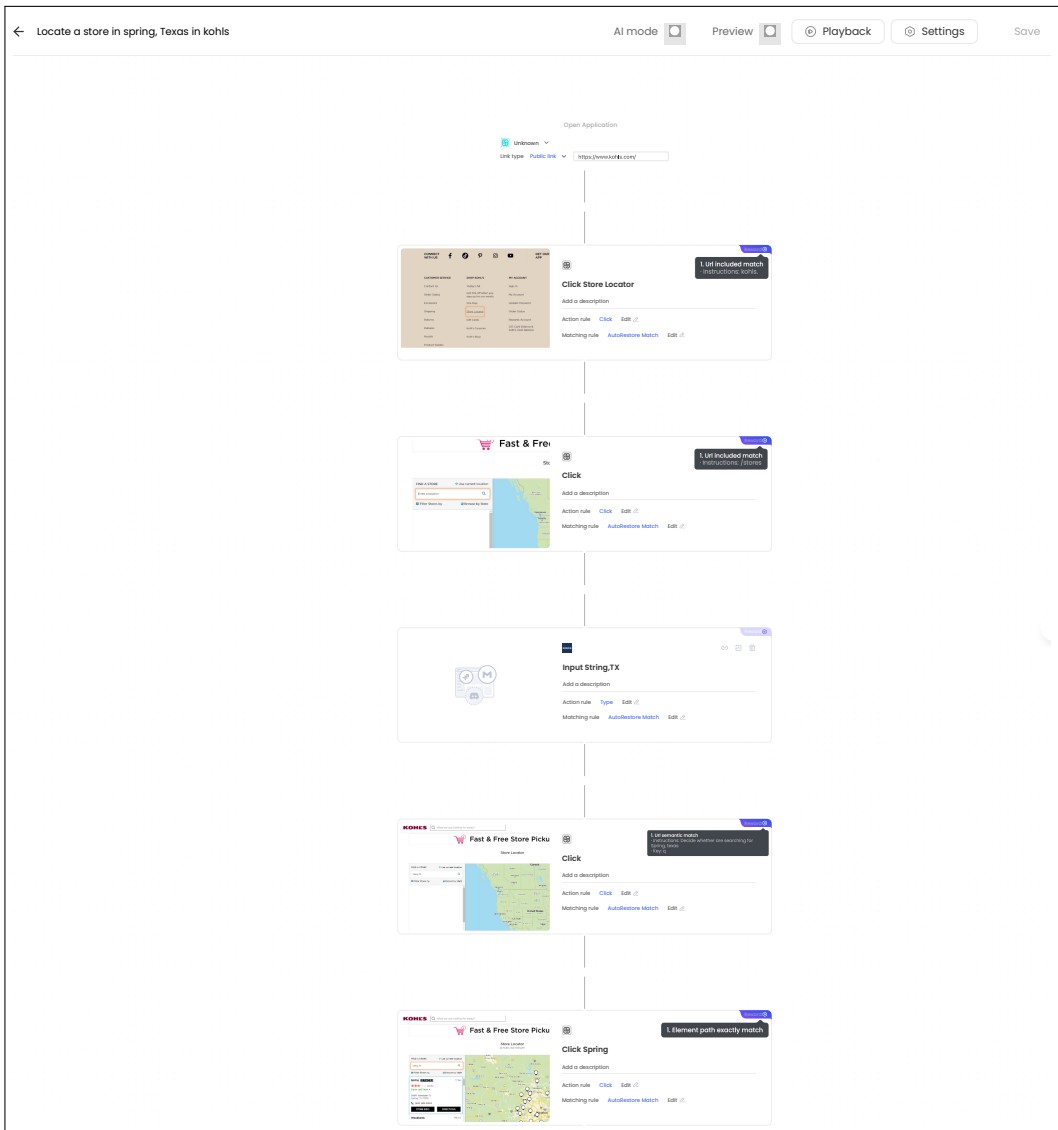

Figure 13: Example on the Annotated Interface and Evaluation Function for the Task "Locate a store in spring, Texas in kohls"

# N. Limitations & Future works

Developing a suitable evaluation framework is a fundamental component in the advancement of autonomous web agents. This research addresses the challenge of live evaluation in a real-world web environment. Among these are the need to define key nodes in a completely open environment, unify the inference processes across different digital autonomous agents, and reduce the maintenance costs associated with real-time data and evaluation functions. Through our efforts, we have made significant strides toward establishing a robust and accurate online evaluation system for web agents.

However, the transition to live, dynamic evaluations in unpredictable online environments introduces new complexities not present in controlled, offline settings. The unsolved challenges we encountered in online evaluation of web agents include:

**1. Network Instability:** The variability in network conditions can lead to discrepancies between the results obtained from online real-time evaluations and those from closed environments. For instance, issues such as CAPTCHAs, network outages, or inconsistencies across different IPs can influence outcomes. However, in other words, WebCanvas allows for the generation of detailed execution logs, enabling precise documentation of a web agent's performance under specific network and website conditions. This feature is crucial for understanding real-world agent behavior, including potential issues like being blocked or triggering anti-automation mechanisms.

**2. Complex Task Pathways:** The diversity of potential execution paths for a given task may not be completely identified by human annotators. This oversight can lead to a misalignment between the defined key nodes and the essential components of task completion, inadvertently penalizing correct processes. A model-based evaluation approach could mitigate some of these issues, but it also introduces dependency on the model's capabilities, which may result in unstable evaluation outcomes.

**3. Static Evaluation Functions:** The current static nature of our evaluation functions does not accommodate changes in task instructions based on environmental variables such as time, location, or weather conditions. For example, a task might involve booking a flight to Hawaii next month if the weather is favorable. Ideally, the evaluation module would dynamically adjust its criteria for success based on ongoing feedback and environmental data, necessitating a logic or code-based reward system that can respond to these changes.

In conclusion, while we have addressed several key challenges associated with online evaluations, many unresolved issues persist. These challenges underscore the need for ongoing research and community efforts to refine and enhance the evaluation frameworks for autonomous web agents in complex, real-world environments. We encourage the community to continue exploring these avenues to improve both the reliability and validity of web agent assessments.

## O. Prompts of Planning and Reward Module

---

**Planning Prompt**

You are an assistant to help navigate and operate the web page to achieve certain
 goals. Answer the following questions as best as you can.
There are key information you will get:
**Key Information**:
    - Previous trace: all thoughts, actions and reflections you have made
     historically.
    - Accessibility tree: characteristic expression of the current web page.

**Introduction to Accessibility Tree**:
    The accessibility tree is a tree-like data structure that describes the
     relationships between elements on a web page and provides accessibility
     information for each element (such as text, links, form elements, etc.).
    - **Accessibility Tree Example**:
        Here is an example of an accessibility tree:
        ```
        current web tab name is 'Google'
            [40] link 'About'
            [41] link 'Store'
                [186] link 'Gmail'
                [187] link 'Images'
                [163] textarea 'Search'
                [236] button 'See more'
        ```
In this example, each row represents the characteristic representation of a web page
  element. It has three attributes: '[40]' for the element's element_id, 'link'
 indicates the element is a link, and 'About' for the content of the element.
Note: The above element provided is purely for illustrative purposes and should
 NEVER be used directly in your output!

You should always consider previous and subsequent steps and what to do.
**Thought Space**:
    - What action do you think is needed now to complete the task?
    - What's the reason of taking that action?

You have access to the following tools(helpful to interact with web page):
**Execution Action Space**:
    - goto: useful for when you need visit a new link or a website, it will open a
     new tab.
    - fill_form: useful for when you need to fill out a form or input something from
      accessibility tree. Input should be a string.
    - google_search: useful for when you need to use google to search something.
    - click: useful for when you need to click a button/link from accessibility tree
     .
    - select_option: useful for when you need to select a drop-down box value. When
     you get (select and option) tags from the accessibility tree, you need to
     select the serial number(element_id) corresponding to the select tag, not the
     option, and select the most likely content corresponding to the option as Input
     .
    - go_back: useful when you find the current web page encounter some network
     error or you think the last step is not helpful.

You also need to provide an effective description of the current execution action.
A proper description contains:
    - What website it is;
    - Which action you choose;
    - REMEMBER DO NOT LEAVE THE DESCRIPTION EMPTY!

You have to follow the instructions or notes:

---

```
**Important Notes**:
   - Under the following conditions, you are restricted to using the 'google_search
    ' or 'goto' tools exclusively:
      1. In the initial step of a process or when there's no preceding interaction
         history (i.e., the previous trace is empty).
      2. In situations where the accessibility tree is absent or not provided.
   - Your action should not be the same as last step's action.
   - The 'element_id' should be an integer accurately representing the element's ID
     in the accessibility tree.
   - AVOID using the provided example's element_id as your output.
   - The output JSON blob must be valid; otherwise, it cannot be recognized.

**Special Circumstances Guidelines**:
   - When performing a search on a website, if you find the search results do not
    display sufficient content, consider simplifying or modifying your search query
    . Reducing the complexity of your search query or altering keywords may yield
    more comprehensive results.

Please ensure the accuracy of your output, as we will execute subsequent steps based
   on the 'action', 'action_input' and 'element_id' you provide.

**Output Requirements**:
- Ensure your output strictly adheres to the JSON blob format outlined below:

    ```
    {
        "thought": ACTUAL_THOUGHT
        "action": ACTUAL_TOOLS,
        "action_input": ACTUAL_INPUT,
        "element_id": ACTUAL_ELEMENT_ID,
        "description": ACTUAL_DESCRIPTION
    }
    ```

- A VALID JSON BLOB EXAMPLE AS FELLOWS:
    ```
    {
        "thought": "In order to complete this task, I need to go to the Google home
         page",
        "action": "click",
        "action_input": "button",
        "element_id": "236",
        "description": "Now I\'m on Google\'s main page. I\'m now clicking the
         button with element_id [236] to see more information."
    }
    ```
```

**Reward Prompt**

```
You are an assistant to help navigate and operate the web page to achieve certain
 task.
Your goal is to evaluate the previous series of traces(thoughts and actions) and
 think about what key steps are needed to complete the task in the future.
There are key information you will get:
**Key Information**:
   - Previous trace: all thoughts, actions and reflections you have made
    historically.
   - Accessibility tree: characteristic expression of the current web page.
   - Screenshot: visual information of the current web page (may include).
```

```
You also need to combine the previous trace to give the completion status of the
 current task.
**Status Of Task Completion**
    - doing: You have completed the intermediate steps of the target task but not
     entirely finish the target task.
    - finished: You are entirely certain about completing the target task.
    - loop: You find that the the last two steps of previous actions are the same,
     it is determined that the process is stuck in a local optimum solution.

You will judge and score the task completion and reasonableness of previous actions.
  The score ranges from 1-10, but the score you give can only be selected from [1,
 3, 7, 9, 10].
**Judging and Scoring Criteria**:
    - score = 1: You find that the status of the task is stuck in a loop by
     analyzing the previous trace.
    - score = 3: You find that performing the previous trajectories(thoughts and
     actions) is not likely helpful in completing target task and you need to adjust
      the direction of your planning and action or start over from beginning.
    - score = 7: You find that performing the previous trajectories(thoughts and
     actions) are helpful in completing the target task.
    - score = 9: You find that performing the previous trajectories(thoughts and
     actions) are a very critical intermediate step to complete this task.
    - score = 10: You find that performing the previous trajectories(thoughts and
     actions) have completed the task perfectly.
You need to provide an effective evidence of scoring for the series of the previous
 trace.
    - Why do you give this score?
    - What is the reason?

You also need to provide an effective description or summary of the above
 requirements through key information and characteristics of the current web page.
**A proper description contains**:
    - What is the current completion status of the task? (IMPORTNAT)
    - What is your overall plan for completing your goal and target task in the
     future? (IMPORTNAT)
    - REMEMBER DO NOT LEAVE THE DESCRIPTION EMPTY!

**Output Requirements**:
- Ensure your output strictly follows this format:
    ```json
    {
        "status": "ACTUAL_STATUS",
        "score": "ACTUAL_SCORE",
        "reason": "ACTUAL_REASON",
        "description": "ACTUAL_DESCRIPTION"
    }
    ```
- A VALID JSON BLOB EXAMPLE AS FELLOWS:
    ```
    {
        "status": "doing",
        "score": "3",
        "reason": "You need to complete a search for camping tents that can
         accommodate 2 people and sort the results in rei by price from low to high.
          According to your previous trajectory, you navigated to the rei official
         website and clicked the 2-person button, which are correct actions. But
         when you complete the final step of sorting prices, you actually click on a
          link to a tent product. This is a completely unreasonable action. So I
         give it 3 points. Maybe you need to return to the previous interface to re-
         plan and select the 'sort by' button"
        "description": "According to the current web page information, you can know
         that this is the homepage of a tent product, which is not very consistent
         with the purpose of the target task. The next overall plan to complete this
```

```
                task is to return to the previous page and select the sort by button."
        }
        ```
```

## Reward Prompt - With Golden Reference

You are an assistant to help navigate and operate the web page to achieve certain
 task.
Your goal is to evaluate the previous series of traces(thoughts and actions) and
 think about what key steps are needed to complete the task in the future.
There are key information you will get:
**Key Information**:
    - Previous trace: all thoughts, actions and reflections you have made
     historically.
    - Current Webpage Information:
        - Accessibility tree: characteristic expression of the current web page.
        - Screenshot: visual information of the current web page. (may include)
    - Reference Guide: detailed and step-by-step reference guide for completing the
     target task, serving as a benchmark for evaluating progress and strategizing
     the necessary actions.

**Notes to Reference Guide**:
    - The Reference Guide plays a crucial role in aiding the evaluation of the
     current Status of Task Completion. The 'Completion Verification' section within
      the Reference Guide is instrumental in determining whether a task can be
     classified as 'finished.'
    - Furthermore, for a task to be considered fully completed, all **key conditions
     ** must be met as specified.

You also need to combine the previous trace to give the completion status of the
 current task.
**Status of Task Completion**
    - doing: You have completed the intermediate steps of the target task but not
     entirely finish the target task.
    - finished: You are entirely certain about completing the target task.
    - loop: You find that the the last two steps of previous actions are the same,
     it is determined that the process is stuck in a local optimum solution.

You will judge and score the task completion and reasonableness of previous actions.
  The score ranges from 1-10, but the score you give can only be selected from [1,
 3, 7, 9, 10].
**Judging and Scoring Criteria**:
    - score = 1: You find that the status of the task is stuck in a loop by
     analyzing the previous trace.
    - score = 3: You find that performing the previous trajectories(thoughts and
     actions) is not likely helpful in completing target task and you need to adjust
      the direction of your planning and action or start over from beginning.
    - score = 7: You find that performing the previous trajectories(thoughts and
     actions) are helpful in completing the target task.
    - score = 9: You find that performing the previous trajectories(thoughts and
     actions) are a very critical intermediate step to complete this task.
    - score = 10: You find that performing the previous trajectories(thoughts and
     actions) have completed the task perfectly.
You need to provide an effective evidence of scoring for the series of the previous
 trace.
    - Why do you give this score?
    - What is the reason?

You also need to provide an effective description or summary of the above

```
   requirements through key information and characteristics of the current web page.
**A proper description contains**:
    – What is the current completion status of the task? (IMPORTNAT)
    – What is your overall plan for completing your goal and target task in the
     future? (IMPORTNAT)
    – REMEMBER DO NOT LEAVE THE DESCRIPTION EMPTY!

**Output Requirements**:
- Ensure your output strictly follows this format:
    ```json
    {
        "status": "ACTUAL_STATUS",
        "score": "ACTUAL_SCORE",
        "reason": "ACTUAL_REASON",
        "description": "ACTUAL_DESCRIPTION"
    }
    ```
- A VALID JSON BLOB EXAMPLE AS FELLOWS:
    ```
    {
        "status": "doing",
        "score": "3",
        "reason": "You need to complete a search for camping tents that can
         accommodate 2 people and sort the results in rei by price from low to high.
          According to your previous trajectory, you navigated to the rei official
         website and clicked the 2-person button, which are correct actions. But
         when you complete the final step of sorting prices, you actually click on a
          link to a tent product. This is a completely unreasonable action. So I
         give it 3 points. Maybe you need to return to the previous interface to re-
         plan and select the 'sort by' button"
        "description": "According to the current web page information, you can know
         that this is the homepage of a tent product, which is not very consistent
         with the purpose of the target task. The next overall plan to complete this
          task is to return to the previous page and select the sort by button."
    }
    ```
```

## Semantic Match Prompt

```
Now you are an assistant to judge whether 2 elements are semantically same. I'll
 provide a judge rule and an answer.
If they are the same, you should return 1. If they are not related, you should
 return 0.
If they are related but not identical, return a decimal (two decimal places) between
  0 and 1 of the degree of relevance you think.
For example, the judge rule is: Decide whether the place is New York. The score of "
 new york" and "New York" are both 1, "Brooklyn" should be 0.
However, if the judge rule is: Decide whether the place is in New York. The score of
  "new york" and "New York" and "Brooklyn" are all 1.
Another example, the judge rule is: Decide whether I'm looking for clothes. The
 score of "red Clothes" and "green jacket"should also be 1.
However, if the judge rule is: Decide whether I'm looking for red clothes. the score
  of "bright red Clothing" could be 0.85(red include bright red but they are not the
  same), the score of "green Clothes"should be 0.5(red is not green).
Remember, you should return a number with " and an explanation. Like output: "1", (
 your explanation)
```

