# OpenReview forum: "WebCanvas: Benchmarking Web Agents in Online Environments"
_ICML.cc/2024/Workshop/Agentic_Markets — Agentic Markets @ ICML'24 Poster_

### Official Review · Reviewer_vFbP · 2024-06-12
**Exciting and scaleable benchmark with promising initial results on building web agents robust to rapidly evolving web environments**

**Rating:** 8
**Confidence:** 4

**Review:**

This paper presents a new way to more comprehensively label tasks on the web and uses it to create a new benchmark by relabeling an existing static state webpage dataset. The proposed method aims to make web agents more robust to dynamic pages and modifications by introducing *key nodes* which effectively break down the trajectories into necessary milestones an agent must traverse to obtain rewards. This work also presents a methodology to maintain an active dataset and a taxonomy on how to classify *key nodes* and successful completion of tasks for semantically challenging settings using an LLM.

The presented labelling method and dataset is complemented by a clear explanation of how to use them in a partially observable Reinforcement Learning setting. This work would benefit by additional clarity in certain sections and more results that capture the full trajectories of the agents across *key nodes* like those in the Appendix J.2 but with particularly difficult or even adversarial examples to fully capture the ability of the dataset to mimic common RL settings.

The authors do not describe the annotation challenges, which can be very relevant to the adaptation of the labelling methodology, and focus on outdated tasks and the tasks discarded due to ambiguity. It is unclear what that ambiguity consists in. Furthermore, it is unclear how the quality of annotation is assessed other than the vague group consensus described in the Appendix B.2. The claim in the conclusion that the platform is “community-driven” is anachronistic as there is no way to judge that at the time of writing. The role of the real active community can only be measured once the platform is made available.

The minor formatting errors the paper could benefit from correcting are mostly related to the order of first mention and explanation for the sake of reading clarity. Element class methods are used to describe *key nodes* that cannot be represented by URLs but it is not explained what Element Class methods are specifically. While Figure 1 is an excellent teaser, Figure 2 has superfluous or unexplained information. The Channel A and B are not mentioned anywhere else and the connection of the top right and bottom right rows are not trivial upon immediate inspection. The header is still the default ICML template one and should be updated with the title of the paper instead.

Overall, the paper is impressive and the results are exciting. I look forward to seeing how this contribution is embraced by the community.

---

### Official Review · Reviewer_MsnA · 2024-06-18
**WebCanvas: Benchmarking Web Agents in Online Environments**

**Rating:** 8
**Confidence:** 4

**Review:**

The paper presents an innovative framework for evaluating web agents in dynamic online settings. Unlike traditional benchmarks that capture static web states, WebCanvas introduces a key-node-based evaluation metric, a refined dataset called Mind2Web-Live, and lightweight annotation tools.

Strengths
Innovative Framework: The introduction of a key-node-based evaluation metric is a notable innovation, addressing the dynamic nature of web interactions.
Comprehensive Dataset: The Mind2Web-Live dataset, which includes 542 tasks and 2439 intermediate evaluation states, provides a robust foundation for benchmarking web agents.
Scalability and Maintenance: The framework supports scalable data recording and annotation, with efficient maintenance strategies to adapt to changes in the web environment.
Detailed Evaluation Metrics: The combination of step scores and task scores allows for a nuanced assessment of agent performance.
Suggestions
Failure Analysis: Can the authors provide a more detailed analysis of the failure cases to better understand the common pitfalls and areas needing improvement?
Key Node Determination: The key nodes are annotated manually. Is there an automatic or semi-automatic to determine the key nodes?
Improvement Strategies: What strategies do the authors propose for improving the task success rates of web agents using the WebCanvas framework?
Recommendation
Rating: 8: Top 50% of accepted papers, clear accept
Confidence: 4: The reviewer is confident but not absolutely certain that the evaluation is correct